



# Predicting power ramps from joint distributions of future wind speeds

Thomas Muschinski[1,2], Moritz N. Lang[1], Georg J. Mayr[2], Jakob W. Messner[3], Achim Zeileis[1], and Thorsten Simon[1,4]

[1]Department of Statistics, Universität Innsbruck, Innsbruck, Austria
[2]Department of Atmospheric and Cryospheric Sciences, Universität Innsbruck, Innsbruck, Austria
[3]MeteoServe Wetterdienst GmbH, Innsbruck, Austria
[4]Department of Mathematics, Universität Innsbruck, Innsbruck, Austria

**Correspondence:** Thomas Muschinski (Thomas.Muschinski@uibk.ac.at)

**Abstract.** Power ramps are sudden changes in turbine power and must be accurately predicted to minimize costly imbalances in the electrical grid. Doing so requires reliable wind speed forecasts, which can be obtained from ensembles of physical numerical weather prediction (NWP) models through statistical postprocessing. Since the probability of a ramp event depends jointly on the wind speed distributions forecasted at multiple future times, these postprocessing methods must not only correct each individual forecast but also estimate the temporal dependencies among them. Typically though, crucial dependencies are adopted directly from the raw ensemble and the postprocessed forecast is limited to the tens of members computationally feasible for an NWP model.

We extend statistical postprocessing to include temporal dependencies using novel multivariate Gaussian regression models that forecast 24-dimensional distributions of next-day hourly wind speeds at three offshore wind farms. The continuous joint distribution forecast is postprocessed from an NWP ensemble using flexible generalized additive models for the components of its mean vector $\mu$ and for parameters defining the forecast error covariance matrix $\Sigma$. Modeling these parameters on predictors which characterize the empirical joint distribution of the NWP ensemble allows forecasts for each hour and their temporal dependencies to be adjusted in one step. Wind speed ensembles of any size can be simulated from the postprocessed joint distribution and transformed into power for computing high-resolution ramp predictions that outperform state-of-the-art reference methods.

## 1 Introduction

Wind power is an environmentally friendly energy source, but difficult to integrate with the electrical grid because of its high temporal variability (Karagali et al., 2013; Sweeney et al., 2020). This temporal variablity stems from its dependence on the hub-height wind speed and is particularly evident when the power produced suddenly changes from very low to very high values, or vice versa. These events are called power ramps and of particular interest to wind farm operators because of their high cost potential (Gallego-Castillo et al., 2015).



There is no single definition for a power ramp – they form a broad class of events with different durations, magnitudes, or types. In all cases though, ramps are joint events which depend on the power – and thus the underlying wind speed – at multiple times. Since the wind speed forecasts for these individual times cannot be assumed to be independent, the true temporal dependencies among them must be estimated in order to reliably predict power ramps (Worsnop et al., 2018). Estimating dependencies between marginal forecast distributions is crucial to many other applications as well, such as wind energy storage sizing (Haessig et al., 2015) or unit commitment (Wang et al., 2011). Typically, multivariate probabilistic power forecasts are issued as scenarios (ensembles) of possible future states (Li et al., 2020).

Predicting the probability that a power ramp occurs during the next day requires reliable wind speed forecasts, which are commonly based on physical numerical weather prediction (NWP) models (Pinson, 2013). NWPs forecast the future state of the atmosphere by numerically integrating governing differential equations in time and space, using current observations from around the world as initial conditions (Richardson, 1922; Bauer et al., 2015). Individual (deterministic) NWPs are not optimal because the atmosphere is a chaotic system and any uncertainties – e.g., in the initial conditions of NWP models – grow rapidly to influence the predictability at future times. Probabilistic wind speed forecasts aiming to quantify this uncertainty are obtained by running an ensemble of individual NWP models, each with slightly different initial conditions or model physics. Since only tens of ensemble members are computationally feasible though, the probabilistic forecast cannot capture the full atmospheric variability and is often underdispersive (Leutbecher and Palmer, 2008).

Improved probabilistic wind speed forecasts that are calibrated – i.e., statistically consistent with observations – and sharp – i.e., with as little uncertainty as possible – can be obtained by postprocessing NWP ensembles using statistical methods (Gneiting et al., 2007). For individual lead times, distributional regressions such as nonhomogeneous Gaussian regression (NGR, Gneiting et al., 2005) or nonhomogeneous regression (Thorarinsdottir and Gneiting, 2010) are commonly employed to forecast continuous (parametric) distributions of wind speed conditionally on the ensemble to correct existing biases or dispersive errors. For multiple lead times, state-of-the-art postprocessing simply reuses separate distributional regressions for the individual lead times without modeling their temporal dependencies (which are crucial for reliable ramp predictions) explicitly. Instead, the separate regressions are combined with ensemble copula coupling (ECC, Schefzik, 2011) to calibrate NWP ensembles for each time, while still retaining the original (temporal) order statistics of their members. This assumption – namely, that dependencies between the postprocessed forecasts are the same as between raw NWP forecasts – is not always fulfilled (Ben Bouallègue et al., 2016) and not supported by the motivation for postprocessing ensembles in the first place. Furthermore, improved forecasts obtained with ECC are again ensembles restricted to the same tens of members and thus not ideal for representing potentially complex multivariate dependencies.

Here we employ novel multivariate Gaussian regression models (MGR, Muschinski et al., 2022) to explicitly include temporal dependencies in the statistical postprocessing of NWP ensembles. MGR is used to forecast wind speeds at 100 m above ground level (AGL) near three offshore wind farms in the North and Baltic Seas. All of the next day's hourly wind speeds are assumed to follow a 24-dimensional Gaussian distribution with mean vector $\mu$ – containing the expectations of the windspeed distributions for each hour – and covariance matrix $\Sigma$ – containing the uncertainties of these individual forecasts as well as their temporal error correlations. The components of $\mu$ and parameters specifying $\Sigma$ are not constant, but estimated conditionally





on the empirical joint distribution of the ensemble using flexible generalized additive models (GAMs) for each distributional parameter. In contrast to state-of-the-art postprocessing methods this forecasts an entire joint distribution conditional on the raw NWP, from which wind speed ensembles of any size can be simulated, allowing higher-resolution power ramp predictions
to be derived.

Ramp probabilities derived from MGR are compared to ECC and other reference methods that forecast a joint distribution of wind speeds – relying either on (i) a multivariate Gaussian distribution fit (GDF, Keune et al., 2014) to the ECC-postprocessed ensemble or (ii) a Gaussian copula estimated from raw observations as motivated by the Schaake Shuffle (Clark et al., 2004) or the prediction errors remaining after univariate postprocessing with NGR (Möller et al., 2013).

Observational data and the NWP ensemble are described in Sect. 2. Postprocessing methods yielding multivariate wind speed forecasts which allow power ramp probabilities to be derived are detailed in Sect. 3. In Sect. 4, the quality of multivariate wind speed forecasts and power ramp predictions are quantified. Finally, the results are discussed in Sect. 5 with some concluding remarks in Sect. 6.

## 2   Data

In order to predict power ramps, numerical weather predictions (NWPs) of wind speed are first postprocessed using the methods presented in Sect. 3 in order to improve their skill. These postprocessing methods model historical observations (Sect. 2.1) on corresponding historical forecasts (Sect. 2.2) using a dataset constructed from the two sources (Sect. 2.3).

### 2.1   Observations from FINO towers

Observations of wind speed are taken from 100 m above ground level (AGL) – the approximate hub height of large offshore
wind turbines – at meteorological towers on the three *Forschungsplattformen in Nord- und Ostsee* (FINO) research platforms. These are located near offshore wind farms in the North Sea (FINO1 and FINO3) and in the Baltic Sea (FINO2). The distributions of wind speeds observed at the three sites are skewed, but observations generally far from zero. Since the NWP ensemble (Sect. 2.2) performs well for next-day lead times, its prediction errors are well approximated by Gaussian distributions. Subsequently, the postprocessing methods described in Sect. 3 – which assume conditional Gaussian distributions for wind speed –
do not require any preliminary transformations to normalize the data.

### 2.2   The ECMWF ensemble

Wind speed forecasts in the period 2016-11-26 and 2021-05-24 are taken from the ensemble system of the *European Centre for Medium-Range Weather Forecasts* (ECMWF). The 50 perturbed ECMWF ensemble members have a spatial resolution of 18 km, with 91 model levels and a temporal resolution of one hour. Forecasts of horizontal wind components $u$ and $v$ at 100
m AGL for hourly lead times between +24h and +47h are bilinearly interpolated to the coordinates of the three FINO towers from which observations are available (Sect. 2.1) The interpolated wind components are used to calculate a wind speed forecast for each ensemble member, lead time, and model initialization.





| Variable | Description |
|---|---|
| $\mathtt{obs}_i$ | Wind speed observed at lead time $i$. |
| $\mathtt{mean}_i$ | Mean of ensemble-member wind speed forecasts at lead time $i$. |
| $\mathtt{logsd}_i$ | Logarithm of standard deviation of ensemble-member forecasts at lead time $i$. |
| $\mathtt{cor}_{ij}$ | (Transformed) correlation between ensemble forecasts at lead times $i$ and $j$. |
| $\mathtt{rho}$ | Average over all $\mathtt{cor}_{ij}$ where $j = i+1$. |
| $\mathtt{yday}$ | Day of year (to capture seasonal variations). |

**Table 1.** Observations and predictor variables used to model distributional parameters for wind speed. The placeholders $i$ and $j$ each stand for one of 24 lead times (+24h, +25h, ..., +47h).

Predictor variables are derived from the ECMWF forecast (Table 1) for each station and model initialization with the goal of characterizing its joint probability forecast. The empirical distribution of the ensemble members for each lead time $i$ is
described by its average $\mathtt{mean}_i$ and log-transformed standard deviation $\mathtt{logsd}_i$. The temporal dependencies between forecasts for individual hours are described by the transformed correlations $\mathtt{cor}_{ij}$, obtained by mapping the interval $(-1,1)$ of the empirical correlations $\rho_{ij}$ to the unrestricted real numbers using the function $r(\rho) = \rho/\sqrt{1-\rho^2}$. It turns out that a first order autoregressive process approximates the temporal dependence of forecast errors quite well, so that $\mathtt{rho}$ – an average of the lag-1 transformed correlations – is included as well. Finally, the day of the year of the ECMWF initialization $\mathtt{yday}$ is added to
account for any seasonal influences that are not captured by the NWP model.

### 2.3 Modeling dataset

For each of the three FINO towers, a dataset is constructed containing values for all variables in Table 1. Each row of the dataset corresponds to one ECMWF initialization, which always occurs at 00 UTC. A single row thus contains the 24 observations $\mathtt{obs}_i$ occuring between 24 and 47 hours after the initialization time, along with the predictors derived from the corresponding
ECMWF forecasts. The datasets have different lengths, with occasional gaps resulting from missing observations. For FINO1 there are 1314 distinct ECMWF initializations available between 2016-11-26 and 2020-08-30 – FINO2 has 1609 initializations between 2016-11-26 and 2021-05-24 and FINO3 1455 initializations between 2016-11-26 and 2020-08-30.

### 3 Methods

Accurately predicting next-day power ramps requires reliable probabilistic wind speed forecasts, which we obtain by statisti-
cally postprocessing NWP ensembles to improve their skill. Forecasts for individual times are commonly postprocessed with distributional regressions, which (i) assume a certain parametric family for the future wind speed observation – such as a Gaussian, truncated Gaussian, or generalized extreme value distribution – and then (ii) model the distributional parameters on predictors derived from the NWP ensemble to address potential biases or miscalibrations (see Lerch and Thorarinsdottir, 2013, for more details and references).





At the FINO stations, hourly wind speed observations $\mathtt{obs}_i$ are approximately normally distributed. Hence a natural first step is to assume that the $\mathtt{obs}_i$ follow separate Gaussian distributions for each of the 24 hours

$$\mathtt{obs}_i \sim \mathcal{N}(\mu_i, \sigma_i), \quad i = +24\mathrm{h}, +25\mathrm{h}, \ldots, +47\mathrm{h}. \tag{1}$$

The mean $\mu_i$ and standard deviation $\sigma_i$ of each distribution are flexibly modeled on the empirical mean and standard deviation of the corresponding NWP ensemble using NGR (Sect. 3.1).

Power ramps depend jointly on wind speeds at different times, so the goal of postprocessing must be a multivariate forecast which not only contains the univariate distributions of Eq. 1, but also characterizes their dependencies. Typically though, multivariate forecasts postprocessed from NWP ensembles are not parametric distributions as in the univariate case, but rather improved ensembles obtained with ECC (Sect. 3.2) that have the same number of members and order statistics as the original NWP, but with hourly forecasts that are corrected according to the distributions predicted by NGR.

For a fully parametric multivariate wind speed forecast, Eq. 1 can be extended to a vector of observations following a multivariate (24-dimensional) Gaussian distribution

$$(\mathtt{obs}_{+24\mathrm{h}}, \ldots, \mathtt{obs}_{+47\mathrm{h}}) \sim \mathcal{N}(\mu, \Sigma), \tag{2}$$

whose $24 \times 24$ covariance matrix $\Sigma$ then contains both the uncertainties (variances) of the hourly forecasts as well as their dependencies (covariances).

The joint distribution in Eq. 2 can be estimated from the ECC-postprocessed ensemble using a multivariate Gaussian distribution fit (GDF, e.g., Keune et al., 2014) as described in Sect. 3.3. This has the advantage of allowing ensembles of any size to be generated through simulation, but as with ECC temporal dependencies are carried over from the original ensemble and cannot be corrected. Alternatively, the 24 hourly distributions postprocessed with NGR can be joined to a single multivariate Gaussian distribution using copulas estimated from observations or forecast errors (e.g., Pinson and Girard, 2012) as described in Sect. 3.4. The problem with this approach is that the physically-based dependencies between NWP ensemble members are not taken into account at all.

Multivariate Gaussian regression (MGR, Sect. 3.5) is a new statistical method developed by Muschinski et al. (2022) which offers a natural solution to these limitations and allows joint distributions of hourly wind speeds to be flexibly postprocessed from NWP ensemble forecasts. MGR naturally extends NGR to multivariate responses, which means that all parameters of the assumed multivariate Gaussian wind speed distribution – not just the marginal forecasts for each hour – can be modeled on predictors derived from the NWP ensemble.

Joint wind speed distribution forecasts are subsequently evaluated using the Dawid-Sebastiani score (Sect. 3.6) and ramp probabilities derived by converting large wind speed ensembles simulated from the distributions into power using an idealized turbine curve (Sect. 3.7).

## 3.1 Nonhomogeneous Gaussian regression (NGR)

Nonhomogeneous Gaussian regression (NGR, Gneiting et al., 2005) offers a way to transform the 50 discrete members predicted by the ECMWF into a more realistic continuous (parametric) wind speed distribution for an individual time, where any



| Model name | Section | Joint distribution forecast | Dependencies are flexibly modeled | Postprocessing in one step |
|---|---|---|---|---|
| *ECC* | 3.2 | | | |
| *GDF* | 3.3 | ✓ | | |
| *GDF (x)* | 3.3 | ✓ | | |
| *COP (Obs)* | 3.4 | ✓ | | |
| *COP (Err)* | 3.4 | ✓ | | |
| *MGR (AR1)* | 3.5 | ✓ | ✓ | ✓ |
| *MGR (AD1)* | 3.5 | ✓ | ✓ | ✓ |
| *MGR (AD2)* | 3.5 | ✓ | ✓ | ✓ |

**Table 2.** Models used to postprocess the multivariate wind speed forecasts, in which section they are introduced, whether or not the postprocessed forecast is a joint probability density function, whether or not dependencies are flexibly modeled, and lastly if postprocessing is a one step procedure or marginal distributions and their dependencies are treated in separate steps.

bias or dispersion error in the original forecast has been corrected. With NGR, the wind speed observed at each hour of the next day is assumed to follow a Gaussian distribution (Eq. 1) whose mean $\mu_i$ and standard deviation $\sigma_i$ are estimated conditionally on predictors that characterize the empirical distribution of the ensemble forecast for that hour.

Following Gneiting et al. (2005), the location and spread of the ensemble are allowed to be corrected differently depending on the time of the year, but rather than employing a sliding window the seasonality is modeled using splines that are estimated from the full dataset as introduced by Lang et al. (2020). Specifically, the mean $\mu_i$ is allowed to have a sesonally varying linear dependence on the mean of the corresponding ensemble forecast $\texttt{mean}_i$

$$\mu_i = f_{0,i}(\texttt{yday}) + f_{1,i}(\texttt{yday}) \cdot \texttt{mean}_i. \tag{3}$$

Analagously, the standard deviation is log-transformed to ensure positivity and modeled on the log-transformed standard deviation of the ensemble $\texttt{logsd}_i$

$$\log(\sigma_i) = g_{0,i}(\texttt{yday}) + g_{1,i}(\texttt{yday}) \cdot \texttt{logsd}_i. \tag{4}$$

The $f$ and $g$ in Eqs. 3 and 4 represent cylical nonlinear yearly effects and are composed of multiple basis functions.

### 3.2 Ensemble copula coupling (ECC)

While NGR can be used to obtain sharp and calibrated forecasts for individual times, dependencies between these forecasts are not considered at all. Ensemble copula coupling (ECC, Schefzik et al., 2013) is often used to generate a multivariate probabilistic forecast by adopting the temporal depencencies of original NWP ensemble members directly, rather than correcting these as is done for the marginal parameters with NGR.

Having postprocssed inividual forecasts with NGR, a cumulative distribution function of the wind speed $\Phi_i$ can be derived for any lead time $i$. Individual ensemble members are then mapped to specific distributional quantiles using the quantile function





$\Phi_i^{-1}$. The result is an ensemble with the same order statistics and number of members as the raw ensemble, but with calibrated margins. Different variants of ECC employ different quantiles for calibrating the margins. Here, quantiles correspond to 50 equidistant probabilities (e.g. $\{1/51, 2/51, \ldots, 49/51, 50/51\}$) and are assigned to the ordered ensemble members.

### 3.3 Multivariate Gaussian distribution fit (GDF)

To avoid the limitations on ensemble size that come with ECC, a joint distribution may be estimated from the postprocessed ensemble for each day using a multivariate Gaussian distribution fit (GDF, Keune et al., 2014). Any number of ensemble members can then be generated by simulating from this distribution. Obtaining the mean vector $\mu$ in Eq. 2 is straightforward – its components are also known from postprocessing the marginal forecasts for each time with NGR.

With the model *GDF*, $\Sigma$ is estimated from the postprocessed ensemble using its sample covariance $S$, where

$$S_{mn} = \frac{1}{N-1} \sum_{i=1}^{N} (x_{im} - \bar{x}_m)(x_{in} - \bar{x}_n), \tag{5}$$

and $x_{im}$ is the wind speed forecasted by ensemble member $i$ for lead time $m$, $\bar{x}_m$ is the ensemble average for lead time $m$, and $N$ is the number of ensemble members ($N = 50$ for the ECMWF ensemble).

Since the number of ensemble members (50) is not much greater than the distributional dimension (24), estimates for
$\Sigma$ can be improved by regularizing with glasso (Friedman et al., 2008). This technique has been used before to postprocess multivariate temperature forecasts (Keune et al., 2014) and in multivariate scoring (Wilks, 2020). The model *GDF ($\delta$)* estimates $\Theta = \Sigma^{-1}$ to maximize the penalized log-likelihood

$$\log(\det \Theta) - \mathrm{tr}(S\Theta) - \delta \|\Theta\|_{\ell 1}, \tag{6}$$

where $\delta$ is a tuning parameter controlling the lasso penalty used to enforce sparsity in the precision matrix $\Sigma^{-1}$.

### 3.4 Gaussian copula (COP)

Alternatively, joint distributions may be generated without using the temporal dependencies from the ECMWF ensemble at all. Instead, a constant correlation matrix – equivalent to a Gaussian copula (Pinson and Girard, 2012) – is estimated and used to join the marginal wind speed distributions for each hour postprocessed with NGR. Following this approach, $\Sigma$ is constructed using a variance-correlation decomposition

$$\Sigma = DPD, \tag{7}$$

where $D$ is a diagonal matrix that contains the standard deviations $\sigma_i$ of the joint distribution known from univariate postprocessing and $P$ is the correlation matrix that specifies temporal dependencies among the individual forecasts.

The model *COP (Obs)* estimates correlations from raw observations as inspired by the Schaake Shuffle. The entries of $P$ are taken to be the sample correlations:

$$r_{mn} = \frac{\sum_{i=1}^{N} (y_{im} - \bar{y}_m)(y_{in} - \bar{y}_n)}{\sqrt{\sum_{i=1}^{N} (y_{im} - \bar{y}_m)^2 (y_{in} - \bar{y}_n)^2}}, \tag{8}$$





where $y_{im}$ is the observation corresponding to model initialization $i$ and forecast hour $m$, $\bar{y}_m$ is the average observation at hour $m$, and $N$ is the total number of model initializations in the dataset (Sect. 2.3).

The model *COP (Err)* takes an analagous approach, but instead estimates P from the errors which remain after univariate postprocessing. These prediction errors take the place of the observations in Eq. 8, so that $y_{im}$ refers to the difference between
the observed wind speed and the mean of the postprocessed univariate Gaussian distribution. In contrast to the other methods discussed so far, *COP (Err)* takes into account the univariate postprocessing step when generating dependencies, but still neglects to use the valuable temporal correlations from the ECMWF.

### 3.5 Multivariate Gaussian regression (MGR)

Multivariate Gaussian regression (MGR, Muschinski et al., 2022) is a novel statistical method that can be used to flexibly
postprocess multivariate ensemble forecasts. It extends nonhomogenous Gaussian regression to joint distributions, so that dependencies between the individual forecasts are also parameterized and modeled instead of being adopted directly from the NWP ensemble, observations or forecast errors. With MGR, observations of next-day hourly wind speeds are assumed to follow a multivariate Gaussian distribution (Eq. 2), whose mean vector $\mu$ and covariance matrix $\Sigma$ are flexibly modeled on predictors characterizing the empirical joint distribution of the ECMWF ensemble. The postprocessed forecast is a continuous
joint distribution estimated uniquely for each ECMWF initialization and can be used to simulate wind speed ensembles of any size.

Modeling the components of $\mu$ is straightforward. These can be linked with the appropriate ensemble means $\texttt{mean}_i$ as in NGR, allowing for seasonally varying bias correction (Eq. 3). On the other hand, $\Sigma$, on becomes more difficult to model beyond the bivariate case employed by Klein et al. (2015) and both Schuhen et al. (2012) and Lang et al. (2019) for modeling
the components of a horizontal wind vector. The reason is that, while for dimension $k = 2$ it is sufficient to ensure that the single estimated correlation parameter has magnitude less than one, higher dimensions require great care to guarantee $\Sigma$ is always positive definite – and thus the joint probability density function well defined – no matter which values the predictors take.

Different parameterizations exist to guarantee positive definite $\Sigma$ (Muschinski et al., 2022). One approach is to use a variance-
correlation parameterization, but this requires assuming errors have a first order autoregressive dependency (Sec. 3.5.1) unless complicated joint constraints are to be enforced. Alternatively, an unconstrained parameterization of $\Sigma$ based on its Cholesky decomposition (Sec. 3.5.2) can also be modeled to allow for more flexible dependencies among the forecasted distributions, but these parameters have the drawback of being somewhat more difficult to interpret.

### 3.5.1 Variance-correlation parameterization

The model *MGR (AR1)* uses a variance-correlation decomposition (Eq. 7) to parameterize $\Sigma$ in terms of the standard deviations $\sigma_i$ contained in $D$ and the correlations in P. Subsequently, the 24 log-transformed standard deviations $\log \sigma_i$ can be modeled as before with NGR (Eq. 4), allowing for seasonally varying dependencies on the log-transformed empirical standard deviations $\texttt{logsd}_i$ calculated from the ensemble.





To ensure $\Sigma$ is always positive definite, a first-order autoregressive (AR1) error dependency is assumed so that the correlation matrix has the form

$$
P = \begin{pmatrix}
1 & \rho & \rho^2 & \cdots & \rho^{23} \\
\rho & 1 & \rho & \cdots & \rho^{22} \\
\rho^2 & \rho & 1 & \cdots & \rho^{21} \\
\vdots & \vdots & \vdots & \ddots & \vdots \\
\rho^{23} & \rho^{22} & \rho^{21} & \cdots & 1
\end{pmatrix}, \tag{9}
$$

and is defined by a single parameter $-1 < \rho < 1$.

In order to model the correlation parameter on predictors, the link function $r(\rho) = \rho/\sqrt{1-\rho^2}$ is used to map its range of possible values to the unrestricted real numbers. The transformed $\rho$ is allowed to have a seasonally varying linear dependence on $\texttt{rho}$, the average of the transformed empirical lag-1 correlations from the ECMWF,

$$
r(\rho) = h_0(\texttt{yday}) + h_1(\texttt{yday}) \cdot \texttt{rho}. \tag{10}
$$

### 3.5.2 Modified Cholesky parameterization

While a variance-correlation parameterization (Sect. 3.5.1) requires joint constraints among the predicted correlations to ensure positive definiteness and a well-defined probability density function, the modified Cholesky parameterization $\Sigma^{-1} = T^{\top} D^{-1} T$, where

$$
D = \begin{pmatrix}
\psi_1 & 0 & 0 & \cdots & 0 \\
0 & \psi_2 & 0 & \cdots & 0 \\
0 & 0 & \psi_3 & \cdots & 0 \\
\vdots & \vdots & \vdots & \ddots & \vdots \\
0 & 0 & 0 & \cdots & \psi_k
\end{pmatrix}, \quad
T^{\top} = \begin{pmatrix}
1 & -\phi_{12} & -\phi_{13} & \cdots & -\phi_{1k} \\
0 & 1 & -\phi_{23} & \cdots & -\phi_{2k} \\
0 & 0 & 1 & \cdots & -\phi_{3k} \\
\vdots & \vdots & \vdots & \ddots & \vdots \\
0 & 0 & 0 & \cdots & 1
\end{pmatrix}, \tag{11}
$$

does not require such constraints. $\Sigma$ is characterized by its innovation variances $\psi_i$ (in $D$, Eq. 11) and generalized autoregressive parameters $\phi_{ij}$ (in $T$), which can be flexibly modeled since positive definiteness is guaranteed for all possible values they may take.

The innovation variances are modeled analagously to the standard deviations in Eq. 4:

$$
\log(\psi_i) = g_{0,i}(\texttt{yday}) + g_{1,i}(\texttt{yday}) \cdot \texttt{logsd}_i. \tag{12}
$$

In applications where the response components have a natural order (e.g., temporal), the high complexity of the regression can be limited by using the autoregressive interpretations of the $\phi_{ij}$ to model a covariance of type order-$r$ antedependence model (AD-$r$). Such a structure for $\Sigma$ can be adopted if forecasts further than $r$ hours apart may be assumed to be conditionally independent. Models are estimated by setting the corresponding $\phi_{ij}$ to zero a priori. For the models *MGR (AD1)* and *MGR*





*(AD2)* this means only modeling the generalized autoregressive parameters with lags at most $r = 1$ and 2, respectively, on empirical correlations from the ensemble $\mathtt{cor}_{ij}$:

$$
\phi_{ij} = \begin{cases} h_{0,i,j}(\mathtt{yday}) + h_{1,i,j}(\mathtt{yday}) \cdot \mathtt{cor}_{ij}, & \text{if } j - i = r \\ 0, & \text{if } j - i > r \end{cases}.
\tag{13}
$$

In contrast to *MGR (AR1)*, these AD-$r$ models are capable of estimating wind speed correlation structures that depend on the
forecast lead time – structures which have been observed by Tastu et al. (2015).

### 3.6   Scoring multivariate wind speed forecasts

Evaluating multivariate forecasts is not straightforward and no score can guarantee optimal forecasts for every application. When evaluating multivariate Gaussian forecasts it is most natural to employ the Dawid-Sebastiani score (DSS, Dawid and Sebastiani, 1999; Gneiting and Raftery, 2007), which is proportional to the log-likelihood of the observations given the post-
processed joint distribution. To ensure a fair evaluation, all wind speed forecasts and subsequently derived ramp predictions – as described in Sect. 3.7 – are scored out of sample using a five-fold cross validation.

### 3.7   Defining and predicting power ramps

Following Worsnop et al. (2018), we predict separate probabilities for up and down ramps of small and large magnitude – i.e., a normalized power change of 0.3 and 0.6 – within three or six hour time spans. To obtain the multivariate power forecasts
required for these predictions, all postprocessed wind speed ensembles – either 50-member ensembles generated by ECC or larger 1000-member ensembles simulated from joint distributions – are first transformed into power space using the theoretical turbine power curve shown in Fig. 3. The probability that a given ramp event occurs is taken to be the fractional number of members which satisfy the ramp criteria – i.e., the number of ensemble members which predict a ramp divided by the total number of members.

Evaluating the skill of these predictions requires observations, which are again obtained through transformations. Here the ramp observation is boolean: either the event occured or it did not. It may happen that no ensemble members predict a ramp on a given day and the estimated ramp probability is zero – especially for the ECC-postprocessed ensemble limited to 50 members. For this reason, ramp forecasts are evaluated using skill scores calculated from the area under the receiver operating characteristic (ROC) curve – the ROCSS – and the Brier score – the BSS – rather than a metric based on the Bernoulli
likelihood.

## 4   Results

The models outlined in Sect. 3 generate joint probabilistic forecasts of the next day's hourly wind speeds from the ECMWF ensemble described in Sect. 2. The improved 24-dimensional forecasts are either (i) ensembles like the original ECMWF prediction or (ii) joint distributions that can be used to simulate any number of possible future wind speed scenarios. The quality of





the joint distribution forecasts is assessed in Sect. 4.1. Subsequently, wind speed ensembles are transformed into power for deriving probabilistic ramp predictions. Since the probability that a ramp event occurs within a certain time window can strongly depend on the correlations between individual wind speed forecasts, it is crucial to accurately model these dependencies in the initial postprocessing step. Results of ramp predictions are presented in Sect. 4.2.

## 4.1   Multivariate wind speed forecasts

Sample postprocessed joint probabilistic wind speed scenarios for 9 Dec 2019 at FINO1 are visualized in Fig. 1. On this wintertime day, observed wind speeds (black lines) decrease throughout the morning before suddenly jumping more than 10 m s$^{-1}$ around noon to cause a significant power ramp which can be seen in Fig. 4 of Sec. 4.2.

  Even without postprocessing, the ECMWF ensemble forecast performs quite well on this day. Observations between 00 UTC and 06 UTC are contained within the narrow ensemble spread (i.e., sharp forecast). All members correctly predict a gradual

decline in wind speed during the morning before a return to higher winds by afternoon. Uncertainty in the ECMWF ensemble increases after 06 UTC because not all ensemble members predict ramps at the same time, but instead up to 6 hours apart.

  While the methods of Sect. 3 all forecast similar marginal distributions of wind speeds, their estimated temporal dependencies are wildly different (Fig. 1). The raw ensemble members have stronger temporal correlations – especially at low lags and during the afternoon – than are forecasted by either *COP (Err)* or *MGR (AR1)*. Regularizing the covariance estimate with

glasso using *GDF (0.5)* weakens the estimated correlations. On the other hand, using correlations from the raw observations with *COP (Obs)* overestimates their strength significantly for all lags at all times of the day – e.g., correlations of more than 0.5 at a high lag of 12h. Subsequently, the forecasted wind speed scenarios appear much too smoth and less noisy than real observations.

  To quantify the forecast quality across all days, postprocessed joint distributions are evaluated using differences in the DSS

(Sect. 3.6) relative to *COP (Err)* and these visualized in Fig. 2. The DSS is quite sensitive to misspecifications in the correlation structure, so that both the unregularized Gaussian distribution fit *GDF* and the observation-based copula *COP (Obs)* have very poor scores – median differences to *COP (Err)* of around -60 and -130, respectively – and must be excluded from the figure. Forecast quality is improved by regularizing *GDF* with glasso – *GDF (0.5)* performs best, while *GDF (0.1)* and *GDF (1.0)* enforce too little and too much sparsity in $\Sigma^{-1}$, respectively – but none match the much simpler reference model *COP (Err)*

which does not use ECMWF dependencies at all.

  The multivariate Gaussian regression model *MGR (AR1)* performs best at each station. Errors are adequately described by a first order autoregressive process and allowing more flexible dependency structures to be modeled using Cholesky-based parameterization – e.g., *MGR (AD1)* or *MGR (AD2)* with assumed first and second order antedependencies, respectively – does not improve the scores.

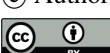

**Figure 1.** Wind speed ensembles at FINO1 forecasted with different postprocessing methods. Forecast members are shown as grey lines and the observations are in black. Where joint distributions are forecasted (bottom two rows), only 50 simulated members are shown for readability. Also included are sample correlations of the ensembles (top row) or the correlations of the joint distributions from which are simulated (bottom two rows).





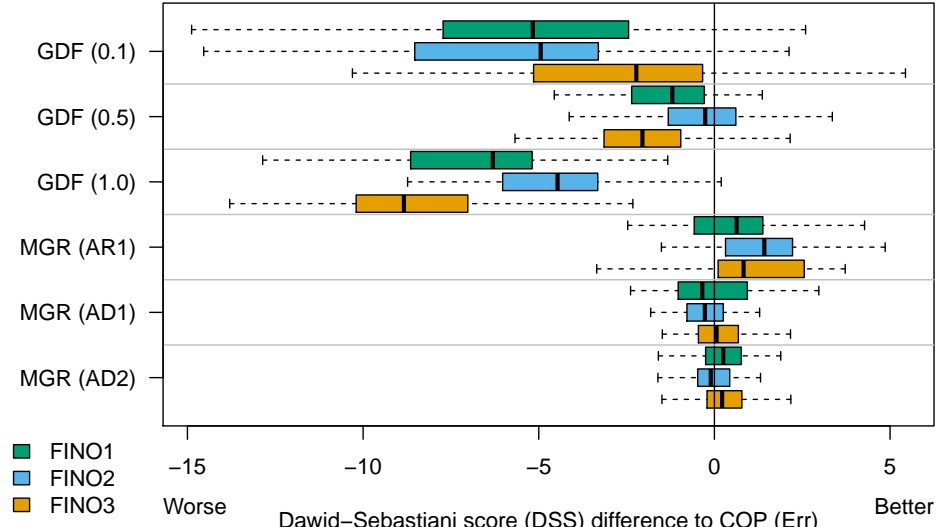

**Figure 2.** Differences in Dawid-Sebastiani Score (DSS) to *COP (Err)*, aggregated by month and year.

## 4.2 Probabilistic power ramp prediction

Postprocessed joint distributions of wind speeds are used to simulate 1000-member ensembles that are converted into multi-variate power scenarios using the theoretical turbine curve of Fig. 3. Scenarios are used to predict probabilities for a wide range of different ramp events as described in Sec. 3.7.

Sample power scenarios at FINO1 are visualized in Fig. 4 for the same day as the wind speeds in Fig. 1. Again only 50 out of 1000 simulated members are shown for readability. Grey lines are the forecasted power scenarios and thick black lines represent the observed power, or more accurately the observed wind speeds transformed into power. A forecast window of width 3 hours beginning at 10 UTC is indicated by shading, during which an up ramp with a magnitude of approximately 0.7 was observed (red colored lines).

Multivariate power scenarios are used to derive probabilities for weak and strong up ramps, where the normalized power increases by 0.3 and 0.6, respectively. These probabilities are included as P(0.3) and P(0.6) in Fig. 4. Both events occured (since 0.7 exceeds 0.3 and 0.6) and *MGR (AR1)* performs the best since it predicts the highest probabilities of them occuring. All other methods predict ramp probabilities that are approximately 40% lower. Furthermore, predictions obtained from *ECC* are also limited to the coarse resolution of the original ECMWF ensemble (1/50), while joint distributions can be used to simulate much larger ensembles for a finer resolution (here 1/1000).

Probabilistic ramp predictions are obtained in this manner for every day and each station. All combinations of ramp magnitudes (weak or strong), types (up or down), forecast window widths (3 h or 6h), and window positions are considered. To quantify the skill of these ramp predictions, probabilities obtained from each postprocessing method are evaluated using the

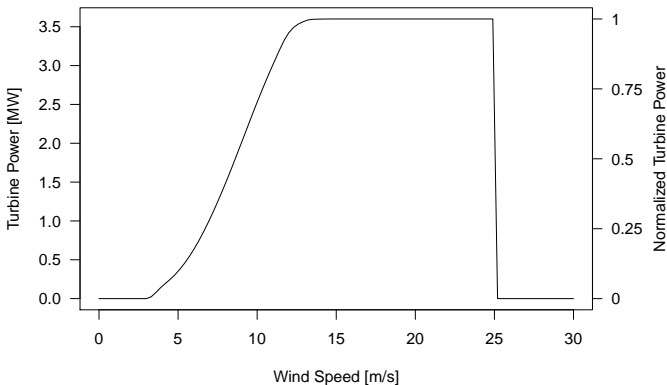

**Figure 3.** Theoretical power curve of a large offshore wind turbine.

scores described in Sect. 3.7 and skill scores computed relative to *ECC* and aggregated over the position of the forecast window
– no significant diurnal variation was observed in the scores.

According to the ROCSS (Fig. 5), all postprocessing methods that predict a joint distribution outperform *ECC* except for
*COP (Obs)* – which has been shown to strongly overestimate temporal dependencies between the invidual wind speed distri-
butions (Sect. 4.1). This improved skill is mainly due to the improved resolution of ramp predictions that result from a larger
number of forecasted power scenarios (1000 vs. 50). The positive influence of resolution on predictive skill is most significant
for rare events – e.g., strong ramps within 3 hours rather than weak ramps within 6 hours.

The BSS relative to *ECC* is visualized in Fig.6. In addition to measuring the resolution of predictions, this score also takes
into account their reliability and the uncertainty of the outcome. As before with the ROCSS, *COP (Obs)* performs worse
than *ECC*, but in contrast not all other joint distribution forecasts have improved skill according to the DSS – e.g., weak
ramps predicted by *GDF (0.5)*. The new multivariate Gaussian regression model *MGR (AR1)* once again performs best overall,
underscoring the importance of statistical methods that can postprocess joint distributions of meteorological quantities from
ensembles and flexibly model their crucial dependencies on data, rather than constructing these through various assumptions.

## 5   Discussion

We use novel multivariate Gaussian regression (MGR) models to postprocess joint distributions of the next day's hourly wind
speeds from NWP ensemble forecasts. Joint distribution forecasts have advantages over traditional ensembles, some of which
are adressed in Sect. 5.1. With MGR, the crucial dependencies linking individual hourly forecasts are parameterized and
modeled explicitly, rather than adopted directly from the ensemble, observations or prediction errors – as is common in state-
of-the-art postprocessing and discussed in Sect. 5.2.

**Figure 4.** The forecasts from Fig. 1, but transformed into power space using the theoretical curve of Fig. 3. Thick black lines are transformed wind speed observations and thin grey lines are transformed wind speed forecast scenarios. A very strong power ramp with magnitude approximately 0.7 (red line) was observed to occur within a specific time window (yellow shading). Power forecasts were used to derive ramp probabilities for up ramps exceeding 0.3 and 0.6 within this time window. They are included as P(0.3) and P(0.6), respectively.



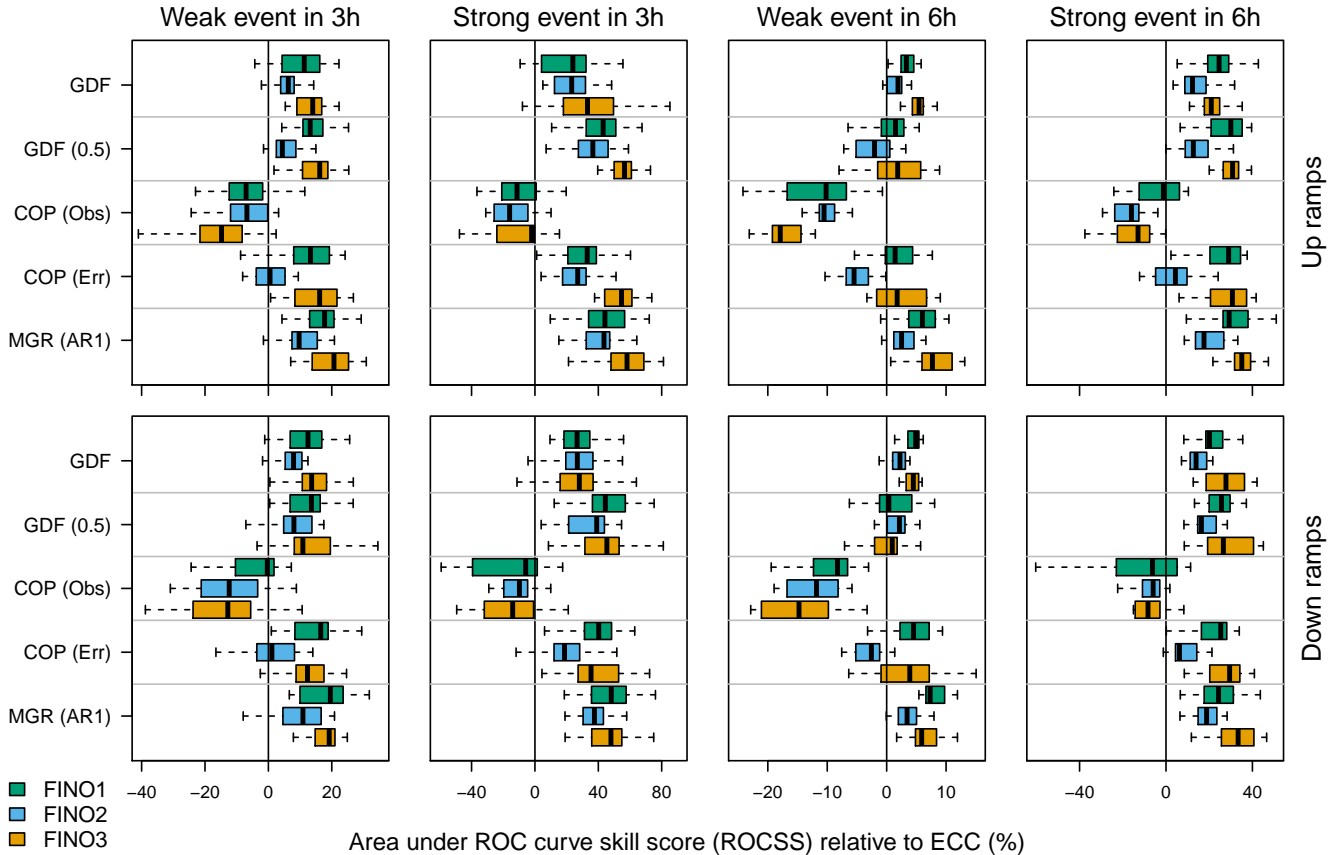

**Figure 5.** Area under ROC curve skill score (ROCSS) relative to *ECC* at the three FINOs for weak and strong up ramps (top row) and down ramps (bottom row) within three and six hour timeframes (leftmost and rightmost two columns, respectively). Individual boxplots are composed of scores computed for specific positions (i.e., starting points) of the time window. Three hour time windows have 22 possible starting points (+24h to +45h) and six hour time frames have 19 possible starting points (+24h to +42h).

## 5.1 Advantages of forecasting a joint distribution

Only tens of ensemble members are computationally feasible for a NWP model. To make the weather forecast more realistic, it is common to generate a continuous distribution from the discrete ensemble members. There are several ways to achieve
this, including (i) replacing individual ensemble members with parametric distributions – e.g., ensemble dressing, Bayesian model averaging – or (ii) modeling a parametric distribution on the ensemble with distributional regression. For multivariate forecasts, there are also similar methods – e.g., ensemble kernel dressing (Schölzel and Hense, 2011), multivariate Gaussian distribution fit (GDF, Keune et al., 2014), and MGR (Muschinski et al., 2022).

Multivariate forecasts postprocessed from NWP ensembles need to accurately describe the true dependencies between fore-
casts at different times. If the result of postprocessing is an ensemble with tens of members, it is poorly suited to accomplish



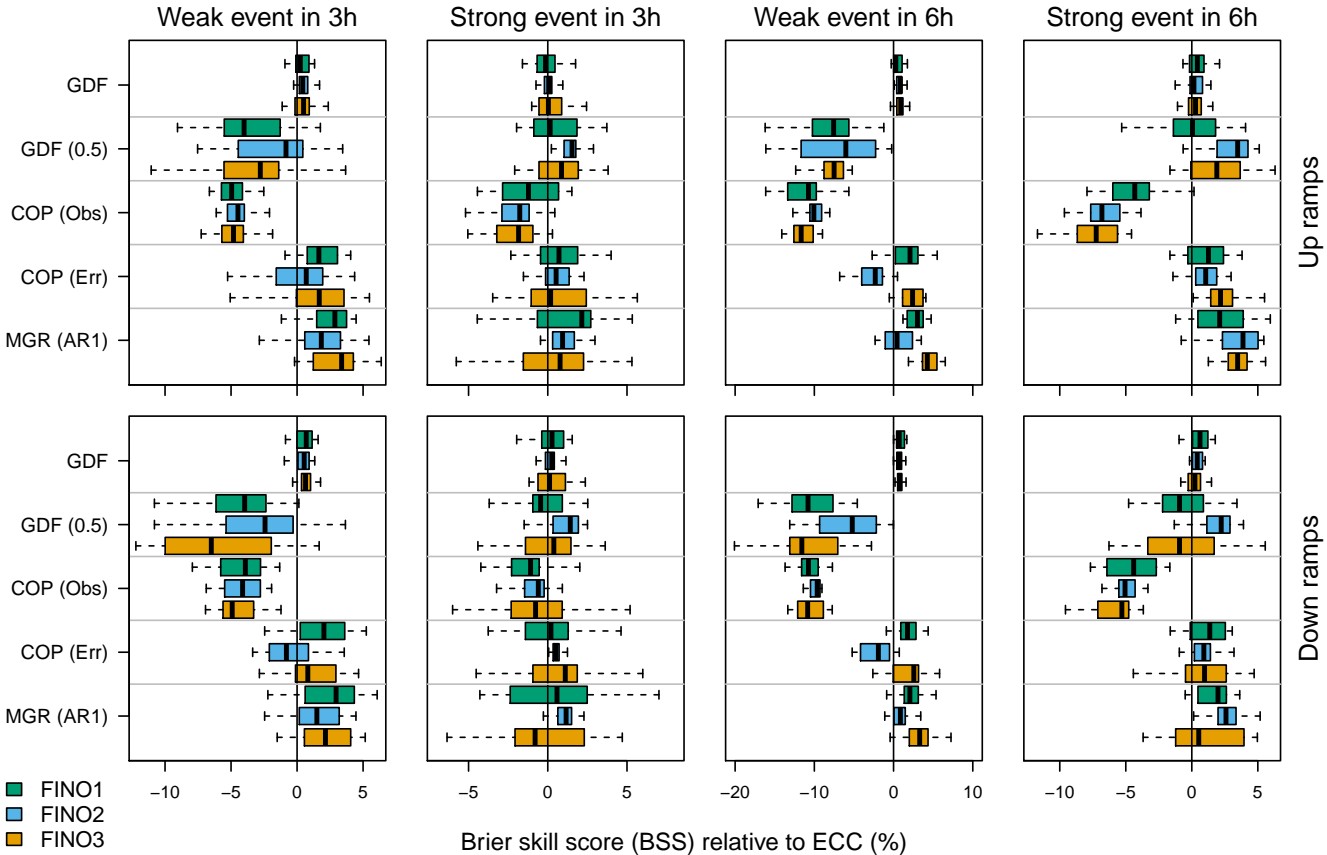

**Figure 6.** As in Fig. 5, but for the Brier skill score (BSS).

this task except for very small dimensions because the complexity of the dependencies increases quadratically with the dimension. At all three FINO stations, ramp predictions are improved when using 1000-member wind speed ensembles simulated from joint distributions instead of 50-member ensembles generated with ECC (Sect. 4.2).

## 5.2 Estimating multivariate dependencies

NWP ensembles are postprocessed to ensure forecasts for any future lead time are calibrated (statistically consistent) and sharp. Since individual ensemble members are distinguishable NWP runs, the dependencies between member forecasts at different times have a physical basis and may contain valuable information regarding the dependencies between the postprocessed forecasts. In state-of-the-art postprocessing, this information is not fully utilized in the same way that the ensemble mean and spread are.

Common methods either directly adopt the order statistics of the NWP ensemble for the dependencies of the postprocessed forecast or apply the same principle using observations. Directly adopting the error dependencies of the NWP ensemble is

most sensible when the raw forecast is already quite good, as is the case here with a forecast horizon of at most 48 hours and a homogeneous ocean surface surrounding the stations. For the same reason, dependencies estimated from observations are much too strong. Presumably these would more accurately reflect the error structure for a very poor NWP prediction, which
approximates an hourly wind speed climatology.

With MGR, a major advantage is the ability to model error dependencies on predictors without a reliance on such assumptions. This allows the temporal error structure of the NWP ensemble to be adjusted for the postprocessed forecast as is commonly done for its location and spread at each time with NGR. If no ensemble is available to obtain the dependencies, MGR can still be used to estimate these from the deterministic forecasts together with external predictors, for example characterizing
the time of the year or the synoptic-scale weather situation.

## 6    Conclusions

Probabilistic power ramp predictions are obtained for three offshore wind farms from joint distributions of hourly wind speeds postprocessed using novel multivariate Gaussian regression (MGR) models. This model employs a multivariate Gaussian distribution for the 24 hourly wind speeds of the next day where the mean vector $\mu$ and covariance matrix $\Sigma$ are estimated
conditionally on an ensemble of numerical weather predictions. This approach provides a couple of advantages compared to current state-of-the-art models that are widely used: First, temporal dependencies between individual hourly forecasts are captured by a flexible regression model rather than assumed to be the same as in the ensemble. Second, forecasts are entire parametric distributions from which samples of any size can be obtained, rather than a sample of a fixed and relatively small size. The latter is particularly important for estimating power ramp probabilities through simulating large ensembles from the
joint wind speed distributions and transforming them into power using an idealized turbine curve. Predictions from MGR outperform state-of-the-art multivariate postprocessing methods according to both the receiver operating characteristic skill score (ROCSS) and the Brier skill score (BSS).

*Author contributions.*  TM, GJM, TS, and AZ planned the research. TM, TS, MNL and AZ developed software. JWM added expertise in wind energy forecasting. TM wrote the original manuscript draft and all authors subsequently reviewed and revised it.

*Competing interests.*  The authors declare that they have no competing interests.

*Acknowledgements.*  This project was funded by the Austrian Science Fund (FWF, grant no. P 31836). We would like to thank the Bundesamt für Seeschifffahrt und Hydrographie for supplying observations from the FINO towers and the European Center for Medium-Range Weather Forecasts for the numerical weather predictions Computational results presented here have been achieved (in part) using the LEO HPC infrastructure of Universität Innsbruck.





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
