# Peer review of "Predicting power ramps from joint distributions of future wind speeds"

_Wind Energy Science, 2022_

## Author Comment (AC2)

**Responses to Reviewer 1**

*The paper describes an improved methodology for the probabilistic forecast of hourly wind speeds on a forecast horizon of one day. Focus of the work is the prediction of power ramps, i.e, the strong increase or decrease of wind power over a time window of one or more hours. Basis of such predictions is typically an ensemble of physics-based numerical wheather predictions (NWP), which is the transformed into wind power predictions using idealized wind turbine power curves.*

*The authors address the temporal multi-point correlation structure of wind speeds and their forecasts as one crucial problem to improve probabilistic forecasts. Together with existing methods they introduce their own approach to explicitly model the joint multivariate distributions of hourly wind speeds with respect to their mutual temporal dependencies. This new approach of Multivariate Gaussian Regression (MGR) has previously been published in a journal on statistics and econometry, and is consequently described only shortly.*

*The main part of the paper performs a detailed comparison of the various methods at the example of one power ramp event measured at the German FINO 1 platform in 2019. The proposed method of MGR outperforms the other appraoches in the comparison.*

*The paper addresses a highly relevant problem in wind power forecasting, namely the so-called power ramps. Moreover, with the temporal multi-point correlation structure of wind speeds the authors address one of the central and most demanding challenges of the field and of atmospheric flows in general. Their approach is promising and the given example is convincing.*

*Technically, the paper is well written and well readable. The structure is clear and comprehensive. English language style is fluent, precise, and, as far as I can say, correct. Results are presented clearly and with very appropriate graphics. References are given whereever necessary.*

*The reviewer is an expert neither in NWP nor in the advanced mathematical approaches of the paper. However, in my eyes this paper makes an important contribution, and it is almost ready for publication.*

Thank you very much for taking the time to read and review our manuscript and for the positive feedback! We are happy to hear you think our paper makes an important contribution. Please find a response to your comments below.

**General remarks**

*The demonstration of the proposed method using just one single example of a power ramp is quite limited. However, given the length of the paper of already 20 pages, more examples do not seem to make sense. Could the authors comment on the performance of the method for more examples, or, elaborate on possibilities of a wider evaluation?*

It is true that we have only visualized wind speed and power ramp forecasts for a single day. On the other hand, the scores we compute and use to quantify forecast performance – of both multivariate wind speed forecasts (via the DSS) and power ramp forecasts (via the area under ROC curve and Brier score) – are based on all forecast days within the dataset and not just the single day visualized.

This has been clarified in lines 67-70 of the revision: *In Sect. 4, the postprocessing methods are first illustrated using an example case where a power ramp occured. Subsequently, the out-of-sample performance of multivariate wind speed forecasts is evaluted across all cases (i.e., days) using scoring rules (Gneiting et al., 2007) and the predictive skill of the ramp probabilities derived from these forecasts quantified as well.*

**Specific remarks**

*P. 3 L. 85: The bi-linear interpolation between grid points is assumably widely used and probably also accepted. However, it is knwon to reduce fluctuation amplitudes. It would be helpful to have any estimate to what extent that effect is present for the given case.*

This is an interesting point! We use a bilinear interpolation because the neighboring grid points of FINO stations are all quite homogeneous (i.e., offshore, over the ocean). Our manuscript is also more focused on investigating

the multivariate postprocessing methods which can be used to predict joint distributions of wind speeds from NWP ensemble forecasts at specific locations, rather than the details of how these ensembles forecasts are obtained in the first place. Still, this is definitely a topic which should be systematically investigated and may play a significant role in certain situations (e.g., perhaps in more complex terrain).

We have touched upon this topic in lines 347-349 of the Discussion: *Although this work has focused on the different methods which can be used to postprocess NWPs, the preprocessing methods initially used to interpolate these NWPs to a specific location also play a role. This could be an interesting topic for future work.*

**Technical remarks**

***P. 3 L. 77:** The phrase "but observations generally far from zero" does not seem to make sense. Please double-check.*

We have rewritten this sentence for improved clarity. It can be found in lines 79-81 of the revision: *The distributions of wind speeds observed at the three sites are skewed, but since the NWP ensemble (Sect. 2.2) performs well for next-day lead times and wind speeds are generally high, prediction errors can be approximated by Gaussian distributions.*

***P. 17 L. 355:** Inserting a "that" after "ensure" would be helpful, although (to my understanding) not strictly necessary.*

We have added a "that" for improved readability.

---

## Author Comment (AC3)

**Responses to Reviewer 2**

*The manuscript, "Predicting power ramps from joint distributions of future wind speeds" describes a novel statistical postprocessing method called the multivariate Gaussian regression (MGR) to provide calibrated and sharp hourly ECMWF 100-m wind speed forecasts for lead times +24h to +47h. Observations from three offshore meteorological towers are used for verification. The main advantages to the new method are that it can explicitly model temporal dependencies across multiple lead times and is not limited to the number of members available in the raw ensemble, unlike the various other multivariate postprocessing methods discussed in the paper and compared against in the figures. Skillful modeling of these joint distributions across lead times is essential for reliable power ramp predictions and consequently the balancing of a power grid with wind energy resources. The authors found that the new method outperformed the various other methods commonly used in the literature when assessing the scores and skill scores of wind speed and wind power forecasts. The paper is overall well-written, interesting, and valuable to the wind energy field. Here are my recommendations for further clarity.*

Thank you very much for taking the time to read our manuscript, for your well written summary, and the positive feedback! You can find our responses to the comments below.

**Scientific and clarification comments**

*L48: I'm not sure what was meant by the phrase, "and not supported by the....." Can you please reword for clarification?*

We have restructured the sentence to clarify our point. It can be found in lines 46 to 48 of the revision: *This assumption – namely, that dependencies between the postprocessed forecasts are the same as between raw NWP forecasts – is not always fulfilled (Ben Bouallègue et al., 2016), for the same reasons that ensemble margins are often miscalibrated and must be postprocessed.*

*L59 and throughout: the use of higher resolution may be confusing to readers in this context as higher resolution when referring to NWP forecasts typically means higher temporal or spatial resolution for numerical integration. Here, it seems to mean more ensemble members. Can you replace with another term or distinguish for the reader the meaning in this paper's context?*

You are right, the multiple meanings of "resolution" could be confusing to the reader. We have added an extra sentence at this point in the text which clarifies the distinction between the resolution of ramp probability forecasts and the resolution of NWP models (in time and/or space). It can be found at lines 60-61 of the revision: *This ramp probability resolution is only a function of the number of ensemble members (i.e., its inverse) and should not be confused with the temporal or spatial resolution of the NWP model.*

*L83-87: Please mention somewhere in here that the forecasts are initialized at 00Z. It's mentioned later, but I found myself asking that question in this section.*

We have added the forecast initalization time as you suggested. It can be found at lines 86-87 of the revision: *The 50 perturbed ECMWF ensemble members have a spatial resolution of 18 km, with 91 model levels and a temporal resolution of one hour and are always initialized at 00 UTC.*

*Sect 3: Wind speeds can only take on positive values. Why not use the truncated Gaussian distribution truncated at zero rather than the untruncated Gaussian distribution?*

In our application we consider wind speeds at 100m AGL over homogeneous (ocean) terrain, where the distributional means are typically large relative to the forecast uncertainty. Subsequently, the large majority of the distribution's mass is positive and taking into account truncation at zero is not so important. We have included Q-Q plots for +24h and +36h forecasts at FINO1 postprocessed with *NGR* below.

**Normal Q–Q Plot**

[Figure]

**Normal Q–Q Plot**

[Figure]

Furthermore, our aim is to forecast turbine power – which is not produced below the cut-in wind speed of 3 m/s – so the exact details of the wind speed distribution around zero are not that important. In the end, a slightly negative wind speed forecast can be treated the same as one which is slightly positive since they both produce no power. This point was already made in Sec. 2.1, but we have now added a quick explanation to the beginning of Sec. 3. It can be found at lines 113-114 of the revision: *At the FINO stations, hourly wind speed observations obs$_i$ are generally large compared to the corresponding spread of the ECMWF ensemble forecast, so that truncation at zero can be neglected.*

***Equations and equation explanations in the text: Some of the equations and explanations of variables in the text reuse symbols which I found difficult to keep track of while reading. For example, i and j referred to lead time and next lead time, respectively in some earlier places in the paper while i referred to ensemble member in other parts (e.g., equation 5) and a new variable, m, became the new lead time variable. The definition of N also changed as well. For better readability, please make the variable symbols unique and consistent throughout the paper.***

We agree that the notation was confusing and have modified the variable names to ensure consistency.

***L253: Can you please provide more details about what the DSS score tells the reader about the forecast? Does it inform about the performance of the calibration and sharpness, just one, or some other aspect of the forecast?***

The DSS is the negative log-likelihood of the observations given the forecast distribution. It is a proper score and thus takes into account both forecast calibration and sharpness. We have added this information in line 260 of the revision: *The DSS is a proper score which measures both sharpness and calibration simultaneously.*

***L295: Can you please state why the COP(Err) was selected as the reference method for wind speed as opposed to the other methods?***

We have chosen *COP(Err)* as the reference method for joint distributional forecasts because it is the existing method which performs best at our stations. According to the DSS, it is only outperformed by (some of the) novel MGR models.

***Figure 1: How were the 50 members out of 1000 selected? Would the gray lines show comparable spread or much more spread if the full 1000 members were shown?***

The 50 members shown in the figure are the first 50 vectors (out of 1000 total) simulated from the postprocessed multivariate Gaussian distribution. An ensemble with 1000 members would be expected to have the same variance (or spread) as one with 50 members since both are simulated from the same distribution. On the other hand, one

would naturally expect the difference between the weakest and strongest wind speeds (i.e., the range of the ensemble forecast) to increase with the number of simulated members.

**Figure 2: The caption says that the data were aggregated by month and year. Is it just showing December data like Figure 1 or all months? If all months, do you expect there to be differences in the scores, and thus optimal method, if all months were not aggregated?**

The boxplots visualize average monthly forecast skill across all months in the dataset (e.g., $12 \cdot 5 = 60$ data points per boxplot for 5 years of forecasts). These values are obtained as follows: First, the DSS for each joint distribution forecast is computed and the difference taken to the DSS of the reference method *COP(Err)*. Subsequently, these differences are averaged for each month of each year (e.g, distinct values for December 2019, January 2020, etc.) and visualized. Alternatively, one could avoid aggregating the scores (i.e., averaging over each month) and instead plot differences in DSS for all forecast days. This does not significantly change the results except that the boxplots span a larger range of values (there is more variance in daily DSS differences than in monthly-averaged DSS differences) and the interpretation of the quartiles changes.

**L323: Can you please state why the ECC method was selected as the reference method for power ramps as opposed to the other methods?**

*ECC* was chosen as the reference method for the power ramp forecasts because it is the standard multivariate NWP postprocessing method. This choice also highlights the improvements in skill that can result from increasing the resolution of ramp predictions by simulating larger ensembles (something which is not possible with *ECC*.)

**Technical comments**

**L148: Spelling of seasonally.**

We've fixed the spelling mistake.

**L160: Spelling of postprocessed individual.**

We've fixed the spelling mistake.

**Eq(5): Please define symbol.**

I'm not sure which symbol in Eq(5) you are referring to. If you mean the $\Theta$ in Eq(6), we have rewritten the equation in such a way that it becomes unnecessary. The variable names in Eq(5) have been changed to ensure consistency following your previous comment.

**L208: I think you can remove on before become.**

Thanks, we have removed the extra *on*.

**Figures 1 & 4: Please put the corresponding forecast hour along the horizontal axis or state the forecast hour range of the figure within the caption.**

We have added the forecast range to the captions of both figures.